# Evaluation of Body-Referenced Graphical Menus in Virtual Environments

Irina Lediaeva*
University of Central Florida

Joseph J. LaViola Jr**
University of Central Florida

## ABSTRACT

Graphical menus have been extensively used in desktop applications and widely adopted and integrated into virtual environments (VEs). However, while desktop menus are well evaluated and established, adopted 2D menus in VEs are still lacking a thorough evaluation. In this paper, we present the results of a comprehensive study on body-referenced graphical menus in a virtual environment. We compare menu placements (spatial, arm, hand, and waist) in conjunction with various shapes (linear and radial) and selection techniques (ray-casting with a controller device, head, and eye gaze). We examine task completion time, error rates, number of target re-entries, and user preference for each condition and provide design recommendations for spatial, arm, hand, and waist graphical menus. Our results indicate that the spatial, hand, and waist menus are significantly faster than the arm menus, and the eye gaze selection technique is more prone to errors and has a significantly higher number of target re-entries than the other selection techniques. Additionally, we found that a significantly higher number of participants ranked the spatial graphical menus as their favorite menu placement and the arm menu as their least favorite one.

**Keywords**: 3D Menus; Menu Placements; Menu Selection Techniques; Menu Shapes; Virtual Reality

**Index Terms**: Human-centered computing—Human computer interaction (HCI)—Interaction paradigms—Virtual reality; Human-centered computing—Interaction design—Interaction design process and methods—User interface design

## 1 INTRODUCTION

Graphical menus are an integral and essential component of user interfaces and have been widely used for 2D desktop applications. Given its wide popularity in desktop applications, graphical menus have been also integrated into virtual environments. Graphical menus in 3D user interfaces are the adapted 2D menus that have proven to be a successful system control technique [16]. For example, Angus and Sowizral [1] developed a hand-held flat panel display that was embedded in a virtual environment and provided a familiar metaphor within the VE context for the users who used graphical menus for desktop applications.

However, once the adapted 2D menus are integrated into 3D space, there are also design challenges that need resolving, such as how best to reach a menu item in 3D space as well as the lack of tactile feedback [9]. Moreover, there are other considerations for designing and implementing graphical menus as system control techniques in 3D user interfaces, such as positioning of graphical user interface elements in space [18], choosing an appropriate selection technique [13], representation of the menu [20] and its overall structure [22].

There are many varieties of graphical menus in virtual environments that have been extensively evaluated by researchers including the TULIP menus [5], spin and ring menus [12], and various menu shapes and menu element sizes [4]. Moreover, Mine et al. [18] have explored body-referenced graphical menus, in which the menu items are fixed to the user's body (and not the head). Such body-referenced graphical menus have several advantages, such as providing "a physical real-world frame of reference, a more direct and precise sense of control, and an 'eyes off' interaction where the users do not have to constantly watch what they are doing." For example, Azai et al. [2,3,4] proposed a menu system that appears at various body parts, such as users' hands, arms, upper legs, and abdomen. However, we found that body-referenced graphical menus are an insufficiently investigated research topic. Such menus are still emerging in the field of virtual environment, and there is a lack of usability studies that would provide design guidelines for developers to implement. This work attempts to close this gap and to provide a comprehensive evaluation of body-referenced graphical menus in virtual environments.

Therefore, the objective of this research is to conduct a comparative study to provide insights and gain a deeper understating of how to design body-referenced graphical menus for virtual environments that support fast completion times, minimize error, and feel natural [5]. During the study, we explored and compared various menu placements, such as placing the menu on the hand [3], arm [2], attaching it to the participant's waist [4] and displaying it in a virtual environment (or in a fixed position in the world) [22]. We examined two menu shapes (linear and radial) [22] and three menu selection techniques (ray-casting [15], head-tracking [21], and eye-tracking [21,23]). Since response time and ease of use of a graphical menu can significantly affect user experience, we gathered typical metrics such as average selection time, the number of wrong selections or error rate and target re-entries, the overall efficiency, user satisfaction and comfort [9].

To evaluate and compare combinations of menu placements, its shapes, and selection techniques, we conducted a user study with 24 participants. We make the following contribution to research on body-referenced graphical menus in virtual environments: design recommendations and insights on user preference of body-referenced graphical menus in virtual environments including recommendations for various menu shapes and selection techniques.

---

* irinalediaeva@knights.ucf.edu
** jjl@cs.ucf.edu

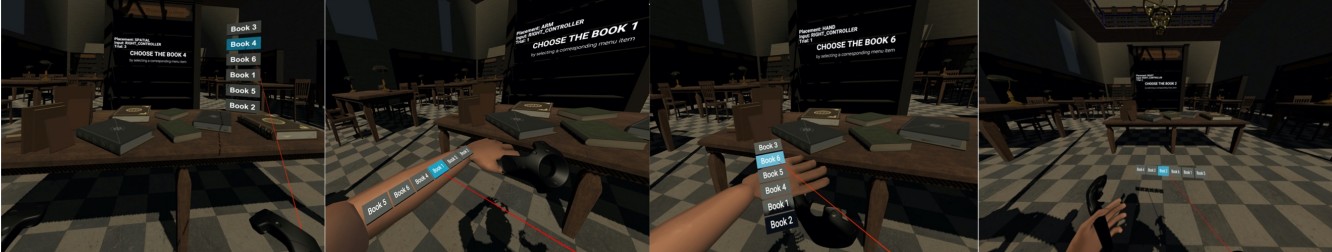

Figure 1: Sample screenshots from the virtual environment. We compared spatial, arm, hand, and waist menu placements (left to right).

## 2 RELATED WORK

Graphical menus in a virtual environment can be classified based on various criteria, such as menu techniques (adapted 2D menus, 1-DOF menus, 3D widgets), placements (world-referenced, head-referenced, body-referenced, etc.), shapes (linear, radial, pie menus, etc.), selections (pointer-directed, gaze-directed, device-directed, etc.). Therefore, the following criteria should be considered when designing graphical menus in a virtual environment: placement, representation and structure (e.g., the spatial layout of the menu, number of menu items, its size, the distance between menu items), and selection [16]. Thus, we situate our study within three main streams of research: 1) menu placements, 2) menu shapes, and 3) menu selection techniques.

### 2.1 Menu Placements

The placement of the menu influences the user's ability to access it (good placement provides a spatial reference) and the amount of menu occlusion in the environment [16]. Feiner et al. [10] first addressed the placement considerations and created a menu taxonomy where the graphical menus can be placed at a fixed location in the virtual world (world-referenced), connected to a virtual object (object-referenced), attached to the user's head (head-referenced) or the rest of the body (body-referenced), or placed in reference to a physical object (device-referenced).

Menu systems that employ the user's body as a graphical menu have been proposed by multiple researchers [2,3,4,18]. For example, Azai et al. proposed a method of displaying the graphical menu in augmented reality on the user's forearm [2] and a menu system that appears at various body parts including not only hands or arms but also the upper legs and abdomen [4]. The researchers found that placing the graphical menu on the body enables the user to operate the menus comfortably and freely [4].

Some research has been done on how to use menus in ways that depart from the typical 2D desktop metaphor. For example, Bowman et al. [6] evaluated the usefulness of letting the participant's fingers perform menu item selection using finger-contact gloves (Pinch Gloves), where the menu items were assigned to different fingers. Another way the menus can be connected to the body is through the body referential zones that are part of an "at-hand" interface [25]. For example, a tool belt surrounding the user may allow them to select objects or options by reaching to a certain location and making a selection. In this study, we particularly focus on investigating body-referenced graphical menus (arm, hand, and waist menus) and comparing it with a conventional world-referenced spatial menu.

### 2.2 Menu Shapes

The items of the graphical menu can be organized in different ways, adopting a radial shape, where the menu items have a circular form, or linear forms, where the menu items have a rectangular form, among other possible configurations (e.g., pie and ring menus).

Researchers have also explored various layouts or shapes of the graphical menus in various environments. For example, Callahan et al. [7] found that menu items in a circular layout perform better in terms of selection time compared to a linear layout in a 2D plane. Similarly, Komerska et al. [17] found that selection using the pie menu for a 3D haptic enhanced environment is considerably faster and more accurate than linear menus. Additionally, Gebhardt et al. [11] presented a formal evaluation of hierarchical pie menus in a virtual environment. Their results indicated high performance and efficient design of this menu type in virtual reality applications. Monteiro et al. [19] found that even though linear and radial menus performed well, the users still preferred the traditional linear menu type and the fixed wall placement of the menu. In this paper we focus on two frequently and widely used types of menus in a virtual environment: linear and radial [22].

### 2.3 Menu Selection Techniques

A menu selection is another form of interaction that is derived from desktop 2D user interfaces. As with desktop menu systems, the user is presented with a list of choices from which they need to select a corresponding menu item. A common selection technique that emulates 2D user interface techniques is where a direction selector or a controller is used to point at, scroll through, highlight, and "click" or trigger a controller button to select various menu items [25].

Researchers have proposed different selection techniques for selecting menu items: ray-casting, head-, eye-, and gesture-based selection techniques. Ray-casting is one of the most well-known menu selection techniques, where a ray is projected from the hand position to the plane of the graphical menu [15]. Further, Qian et al. [21] investigated eye-based and head-based selection techniques and concluded that the eye-only selection offered the worst performance in terms of error rate and selection times.

To best of our knowledge, this study is the first to systematically explore graphical menus in a virtual environment with relevance to menu placements, shapes, and selection techniques. The main contribution of this study is to provide a set of design guidelines for developing body-referenced graphical menus in virtual environments.

## 3 USER STUDY

We conducted a user study to evaluate a variety of graphical menus placements (spatial, arm, hand, waist) in a virtual environment (**Figure 1-2**) as well as its menu shapes (linear and radial), and selection techniques (ray-casting with a controller device, head, and eye gaze). The following sections describe the design, tasks, and measurements.

### 3.1 Study Design

This within-subjects study consisted of 3 independent variables: menu placement (spatial, arm, hand, waist), menu shape (linear and

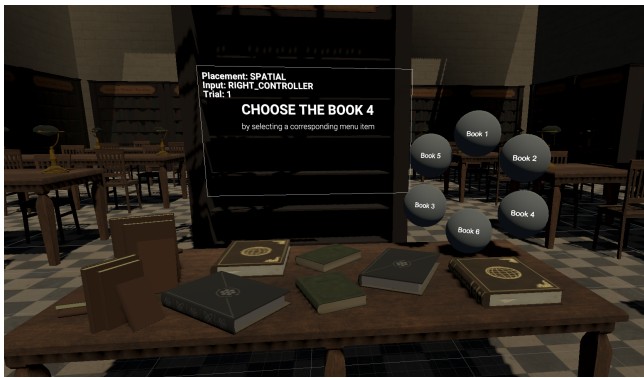

Figure 2: Sample screenshot from the virtual environment with a radial graphical menu (spatial menu placement).

radial), and menu selection technique (ray-casting with a controller device, head, and eye gaze). In total, we had $4 \times 2 \times 3 = 24$ conditions and for each condition, the participant conducted 10 trials which make a total of 240 selections per participant as part of the user study. Each condition was presented to the user in a random order based on a Latin square for constructing "Williams designs" [24]. For each condition, users were asked to select 10 randomly generated items (one item at a time).

### 3.2 Dependent Variables

Our dependent variables were average task completion time, where the average is taken over the 10 trials for that condition, error rates, number of target re-entries, and post-questionnaire responses. We automatically recorded our dependent variables throughout the whole study session and stored the data in a text file. Task completion time (TCT) was measured as the time from the moment the system displayed a message to the moment the user selected a menu item. Error rates (ERR) were recorded every time the user pressed the wrong menu item which was different than the system message requested (the percentage of wrong selections for a given condition). Number of target re-entries (TRE) was measured as a number of times the pointer left the volume of the target menu item and then went again inside the target.

### 3.3 Study Hypotheses

Based on the related work [7,19,21] and our pilot studies during the design of our experiment, we devised the following hypotheses for our study:

- **Hypothesis 1 (H1):** Spatial graphical menus in conjunction with the radial shape and the ray-casting selection technique will let users perform the menu tasks faster, make less errors and take a smaller number of target re-entries in comparison with the other types of graphical menus in a virtual environment.
- **Hypothesis 2 (H2):** Participants will prefer to use spatial graphical menus in conjunction with the radial shape and the ray-casting selection technique than the other types of graphical menus in a virtual environment.

### 3.4 Tasks

In the study, our participants were immersed in a virtual environment that portrayed a virtual library in which the user needed to select books. We had a total of 24 conditions in our study. Each condition consisted of 10 tasks. In these tasks, the participant was asked to select a graphical menu item using the given selection technique. We used 6 menu items for each type of the menu (we found this number appropriate for a menu), and all the menu items were labeled with numbers from 1 to 6 in our experiment. The menu items were shuffled for each condition to prevent a learning effect.

For each menu task, the book number was randomly generated and automatically displayed in a system message (e.g., "Choose Book 4"). For each menu condition, the system message was placed near the center of the virtual table with books. The system message also included information about the current trial, menu placement, and selection technique that were automatically generated by the controller algorithm (to account for all conditions). In other words, once the participant performed a menu task, the system automatically generated and placed a new menu condition in the virtual environment and displayed that information to the participant with a book number in the system message. The participant had a 2-second break between the menu item selection tasks.

For the selection techniques, the participant used a controller device (with a ray cast), a head movement (with head-tracking), or eye-tracking to point at a menu element; then they needed to press the controller trigger button to select the corresponding menu item. Furthermore, for each selection technique, the participant could see a bullet mark that was highlighted (e.g., at the end of the ray cast) once the participant navigated toward the graphical menu. That way the participant could easily select the menu items using all the aforementioned selection techniques. The laser selection pointer with a bullet mark was only displayed for a ray cast selection technique. For the eye gaze and head selection techniques, only a bullet mark was visible.

The participant was informed whether a selection was right or wrong by a sound alert. If the selection was right, the participant could proceed to the next menu task. This process continued until all the menu combinations were completed.

### 3.5 Participants and Apparatus

We recruited 24 participants (20 males and 4 females) between the ages of 18 to 32 years old ($\mu = 21.54$, $\sigma = 4.06$), of which two participants were left-handed and one was ambidextrous. 16 participants had a high school diploma, 2 participants had an associate degree, and 6 participants had a bachelor's degree as their highest completed level of education. 11 participants identified their ethnicity as White/Caucasian, 6 participants were Hispanic/Latino, 5 participants were Asian/Pacific Islander, and 2 participants were Multiethnic/Other. Furthermore, 11 participants specified that they wore glasses or contacts. A Likert scale from 1 to 7 with 1 representing little experience or familiarity and 7 representing very experienced or greater familiarity was used to measure the following in a demographic survey: participants experience with VR ($\mu = 4.46$, $\sigma = 1.63$), familiarity with game controllers ($\mu = 5.75$, $\sigma = 1.66$), eye-tracking technology ($\mu = 3.67$, $\sigma = 1.84$), and video games ($\mu = 5.42$, $\sigma = 1.6$).

The experiment duration ranged from 30 to 45 minutes and all participants were paid $10 for their time. The experiment setup consisted of a 55" Sony HDTV for the experimenter to view, a Tobii HTC Vive headset, which is a retrofitted version of the HTC Vive headset with complete eye-tracking integration from Tobii Pro, Vive controllers, and a Vive tracker attached to a TrackBelt band in order to track the position of the user's waist. We used the Unity game engine for implementing all the graphical menus in conjunction with Tobii XR SDK[1] for integrating eye-tracking

---

[1] https://vr.tobii.com/developer/

technology for the eye gaze selection technique, and FinalIK[2] for tracking the full body of the user and placing the graphical menus on the user body. The software was run on a Windows 10 desktop computer in a lab setting, equipped with an Intel Core i7-4790K CPU, an NVIDIA GeForce GTX 1080 GPU, and 16 Gigabytes (GB) of RAM.

Table 1: Post-questionnaire. Questions 1-7 were asked to rate each placement menu (spatial, arm, hand, waist) on a 7-point Likert scale, questions 8 and 9 were multiple-choice questions, and the question 10 was open-ended.

| Q1 | To what extent did you like the placement menus? |
|---|---|
| Q2 | How mentally demanding were the placement menus? |
| Q3 | How physically demanding were the placement menus? |
| Q4 | How successfully were you able to choose the menu items you were asked to select? |
| Q5 | Did you feel that you were trying your best? |
| Q6 | To what extent did you feel frustrated using the placement menus? |
| Q7 | To what extent did you feel that the placement menus were hard to use? |
| Q8 | Which shape of menu would you prefer for the menu placements in VR? Linear, radial, or all equally? |
| Q9 | Which selection technique would you prefer for the menu placements? Controller, head, eye gaze, or all equally? |
| Q10 | What are your further comments on your experience with the graphical menus in VR? |

## 3.6 Study Procedure

The study started with the participant seated in front of the TV display and the experimenter seated to the side. Participants were given a consent form that explained the study procedure and the participant's rights and responsibilities. They were then given a demographic survey that collected general information about the participant age, gender, ethnicity, dexterity, familiarity with virtual reality, game controllers, eye-tracking technology, and how often the participants played video games. After that, the participant was guided on how to position their headset and adjust the interpupillary distance (to align the lenses with the distance between the participant's pupils) for the best visual experience followed by a quick 5-point calibration to adjust the eye tracker.

During the session, participants were only seated (they did not stand or walk around) with the headset on, an attached Vive tracker to their waist, and two controllers in their hands for selecting the menu items. The participants' limbs were at rest on their side. The participants were asked to comfortably lean on the back of a chair and be in a relaxed position. The participants' limbs were always in the necessary starting position to trigger the menus. The position of the arm and hand menu placements were changed either to the left or right of the participant as well as the trigger selection method (right or left controller) based on the participant's dexterity (left-handed or right-handed) in order to make sure the participant felt comfortable while completing the menu tasks.

Next, the participant conducted a training session of 5-10 minutes in order to get familiar with the virtual environment and the different types of graphical menus including menu placements and selection techniques. During the training session, each menu combination was completed in 5 trials. The system displayed a

[2] http://www.root-motion.com/final-ik.html

message with the number of the book from 1 to 6 selected randomly. The participant needed to select the corresponding item from the menu.

Once the training session was completed and the participant felt comfortable with the selection techniques and menu placements, the participant started the study session to further evaluate various combinations of the graphical menus. At the end, the participant filled out a post-questionnaire (**Table 1**) using a 7-point Likert scale (e.g., from 1 or "not at all mentally demanding" to 7 or "extremely mentally demanding") for each placement menu (spatial, arm, hand, waist) and ranked them based on the overall preference, how mentally or physically demanding the placement menus were, frustration, and its ease of use. Additionally, we asked the participant to select preferred menu shapes and selection techniques for each placement menu and leave additional comments on experience with the body-referenced graphical menus.

## 4 RESULTS

### 4.1 Quantitative Results

We used a repeated measures ANOVA per each dependent variable. For the ANOVA results that are significant, we performed pairwise sample t-tests to see what conditions are specifically significant. We used Holm's sequential Bonferroni adjustment to correct for type I errors [14]. **Table 2** shows the results of repeated measures ANOVA analysis.

Table 2: Repeated measures ANOVA results for comparing menu placement, shape, and selection technique.

| Source | Task Completion Time | Error Rates | Number of Target Re-Entries |
|---|---|---|---|
| Placement | $F_{3,69} = 13.735$ $p < .0005$ | $F_{3,69} = 3.41$ $p < .05$ | $F_{3,69} = 0.393$ $p = .758$ |
| Shape | $F_{1,23} = 1.078$ $p = .31$ | $F_{1,23} = 2.448$ $p = .131$ | $F_{1,23} = 2.217$ $p = .15$ |
| Selection | $F_{2,46} = 6.562$ $p < .005$ | $F_{2,46} = 9.741$ $p < .0005$ | $F_{2,46} = 39.075$ $p < .0005$ |
| Placement × Shape | $F_{3,69} = 2.881$ $p < .05$ | $F_{3,69} = 2.065$ $p = .113$ | $F_{3,69} = 2.889$ $p < .05$ |
| Shape × Selection | $F_{2,46} = 1.642$ $p = .205$ | $F_{2,46} = 0.443$ $p = .645$ | $F_{2,46} = 1.947$ $p = .154$ |
| Placement × Selection | $F_{6,138} = 2.47$ $p < .05$ | $F_{6,138} = 1.921$ $p = .082$ | $F_{6,138} = 2.485$ $p < .05$ |
| Placement × Shape × Selection | $F_{6,138} = 1.706$ $p = .124$ | $F_{6,138} = 2.469$ $p < .05$ | $F_{6,138} = 0.459$ $p = .837$ |

#### 4.1.1 Main Effect of Placement

We found significant difference in average task completion time ($F_{3,69} = 13.735$, $p < .0005$) and error rates ($F_{3,69} = 3.41$, $p < .05$) between menu placements.

**Task Completion Time:** Participants took significantly longer to complete the menu tasks when the graphical menu was placed on the arm than when the graphical menu was placed spatially ($t_{23} = -5.253$, $p < .0005$), on the waist ($t_{23} = 4.59$, $p < .0005$) or on the hand ($t_{23} = 4$, $p < .005$).

**Error Rates:** Participants made significantly more errors when using the hand placement graphical menus than the arm graphical menus ($t_{23}$ = −4.079, $p$ < .005).

### 4.1.2 Main Effect of Shape

Menu shapes did not have any significant effect on task completion time ($F_{1,23}$ = 1.078, $p$ = .31), error rates ($F_{1,23}$ = 2.448, $p$ = .131) or number of target re-entries ($F_{1,23}$ = 2.217, $p$ = .15).

### 4.1.3 Main Effect of Selection Technique

We found significant difference in average task completion time ($F_{2,46}$ = 6.562, $p$ < .005), error rates ($F_{2,46}$ = 9.741, $p$ < .0005), and number of target re-entries ($F_{2,46}$ = 39.075, $p$ < .0005). between selection techniques.

**Task Completion Time:** Further, we found that the head selection technique took significantly more time to complete a menu task across different placements than the ray-casting selection technique ($t_{23}$ = −7.238, $p$ < .0005).

**Error Rates:** Participants made significantly more errors when using the eye gaze selection technique than other selection techniques, such as the head ($t_{23}$ = −3.868, $p$ < .005) or ray-casting ($t_{23}$ = −2.219, $p$ < .005) techniques. Likewise, participants made significantly more errors using the ray-casting selection technique than using the head selection technique ($t_{23}$ = 3.391, $p$ < .05).

**Number of Target Re-Entries:** The eye gaze selection technique had a significantly higher number of target re-entries than the ray-casting ($t_{23}$ = −5.864, $p$ < .0005) or head ($t_{23}$ = −6.688, $p$ < .0005) selection techniques. Also, we found the ray-casting was significantly higher in terms of number of target re-entries in comparison with the head selection technique ($t_{23}$ = 3.008, $p$ < .005).

### 4.1.4 Interaction Effect of Placement × Shape

We found significant differences in average task completion time ($F_{3,69}$ = 2.881, $p$ < .05) and number of target re-entries ($F_{3,69}$ = 2.889, $p$ < .05) between menu placements and shapes.

**Task Completion Time:** We found that the arm placement menu in conjunction with the linear shape took significantly more time to complete a menu task than the linear hand ($t_{23}$ = 4.994, $p$ < .0005), spatial ($t_{23}$ = 4.558, $p$ < .0005), waist ($t_{23}$ = 3.573, $p$ < .01) and the radial spatial ($t_{23}$ = 5.481, $p$ < .0005) and waist ($t_{23}$ = 4.533, $p$ < .0005) graphical menus. Additionally, the arm menu placement with the radial shape took more time than the linear spatial menu ($t_{23}$ = 3.787, $p$ < .01) and radial spatial ($t_{23}$ = 3.494, $p$ < .01) and waist ($t_{23}$ = 3.476, $p$ < .01) menus.

**Number of Target Re-Entries:** The linear arm menu had a significantly higher number of target re-entries than the radial arm placement menu ($t_{23}$ = 3.726, $p$ < .01).

### 4.1.5 Interaction Effect of Shape × Selection

We did not find any significant effect on task completion time ($F_{2,46}$ = 1.642, $p$ = .205), error rates ($F_{2,46}$ = 0.443, $p$ = .645) or number of target re-entries ($F_{2,46}$ = 1.947, $p$ = .154) between menu shapes and selection techniques.

### 4.1.6 Interaction Effect of Placement × Selection

We found significant differences in average task completion time ($F_{6,138}$ = 2.47, $p$ < .05) and number of target re-entries ($F_{6,138}$ = 2.485, $p$ < .05) between menu placements and selection techniques.

**Task Completion Time:** For the hand placement menus, we found that it takes significantly more time for participants to complete a menu task using the head selection technique than ray-casting ($t_{23}$ = -6.06, $p$ < .0005). For spatial ($t_{23}$ = -5.102, $p$ < .0005)

and waist ($t_{23}$ = -4.648, $p$ < .0005) menus, again the head selection technique performed poorly than the ray-casting technique.

**Number of Target Re-Entries:** We noticed that the eye-gazing selection had a higher number of target re-entries for each placement. For example, the arm ($t_{23}$ = -5.383, $p$ < .0005), waist ($t_{23}$ = -6.263, $p$ < .0005), spatial ($t_{23}$ = -4.045, $p$ < .01) eye-gaze menus had significantly more re-entries than the arm, waist, and spatial head selection technique. Furthermore, the hand ($t_{23}$ = -7.233, $p$ < .0005), waist ($t_{23}$ = -6.042, $p$ < .0005), and spatial ($t_{23}$ = -3.82, $p$ < .01) eye gaze menus performed poorly in terms of re-entries than the hand, waist, and spatial ray-casting graphical menus.

### 4.1.7 Interaction Effect of Placement × Shape × Selection

We found that menu placements in conjunction with menu shapes and selection techniques had a significant effect on error rates ($F_{6,138}$ = 2.469, $p$ < .05). Additionally, based on our hypotheses, we did more investigations to see if the spatial graphical menus in conjunction with the radial shape and the ray-casting selection technique are indeed better than the other combinations of the graphical menus in terms of error rates. We found that there was no significant effect in error rates between the spatial radial ray-casting menu and other combinations of graphical menus.

## 4.2 Qualitative Results

We did the Chi-squared test for the gathered post-questionnaire data about preferred menu shapes and selection techniques. We did not find any significant difference for the spatial and arm menus in terms of menu shapes. However, we found significance for the hand $X^2$ (2, $N$ = 24) = 12.5, $p$ < .05 and waist $X^2$ (2, $N$ = 24) = 7.605, $p$ < .05 menus. Moreover, 12 people thought that all shapes were equivalent for the spatial graphical menus, 9 and 8 participants preferred the linear and radial shapes respectively for the arm graphical menus, 14 participants preferred the radial shape for the hand graphical menus, and 13 participants thought that the linear shape is a good fit for the waist menu (**Figure 3**).

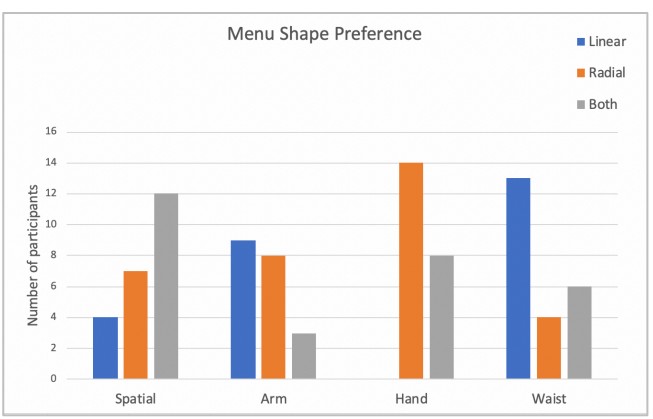

Figure 3: Menu shape preference for each placement menu.

For the selection techniques, we found significant difference for the spatial $X^2$ (6, $N$ = 24) = 22.66, $p$ < .05, arm $X^2$ (6, $N$ = 24) = 19.62, $p$ < .05, and waist $X^2$ (6, $N$ = 24) = 19.12, $p$ < .05 menu placements. Also, participants thought that all selection techniques were equivalent for the spatial graphical menus, eye gaze was a good fit for the arm menus, eye gaze or ray-casting selection techniques were overall better for a hand menu, and eye gaze was a favorite selection technique for the waist menu (**Figure 4**). The spatial graphical menu was ranked as the overall best menu placement and the arm graphical menu as the worst placement. The

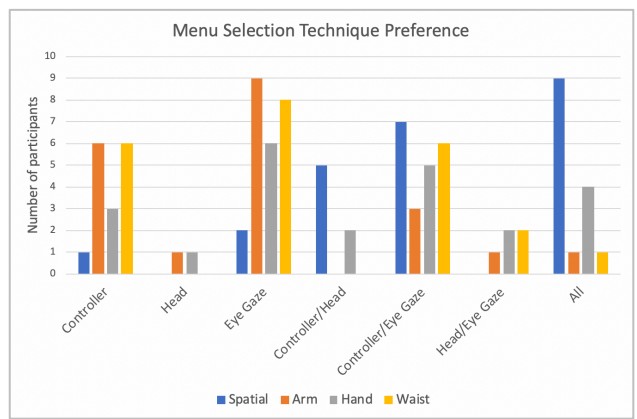

Figure 4: Selection technique preference for each placement menu.

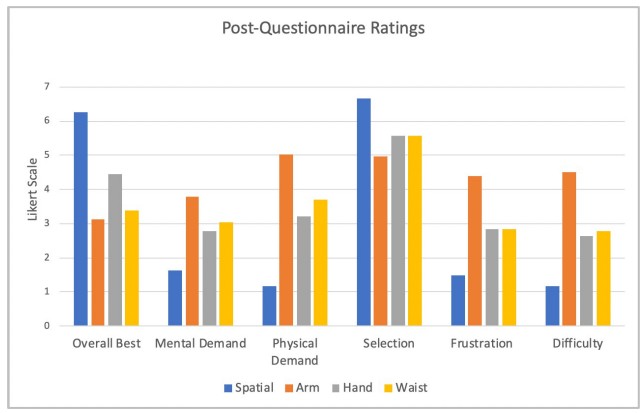

Figure 5: Post-questionnaire ratings for each placement menu.

spatial graphical menu was also ranked as best in terms of ease of use, mental and physical demand, and selection rates (**Figure 5**).

To analyze the Likert scale data, we used Friedman's test followed by a post-hoc analysis using Wilcoxon signed-rank tests for pairs (**Table 3**). Average ratings for post-questionnaire questions 1 to 7 are summarized in **Figure 5**. From the results that we obtained we concluded the following:

- Participants liked the spatial menus compared to the arm, hand, and waist graphical menus. The arm menu was the participant's least favorite.
- Spatial and hand graphical menus are less mentally demanding compared to the arm and waist graphical menus.
- Spatial menus were significantly less physically demanding compared to the hand, arm, and waist graphical menus. The arm menu was the most physically difficult.
- Participants were able to successfully select the menu items from all the types of placement graphical menus.
- The frustration level was higher for the arm graphical menu and significantly lower for the spatial graphical menu.
- Participants thought that the arm graphical menu was significantly harder to use than the other types of placement menus.

Based on the comments that we gathered from the post-questionnaire, we found the following emerging themes. 6 of the participants reported that the arm menu was in an uncomfortable position, was difficult and tiresome to use: *"I found the arm slightly more physically demanding simply because it required me to move my neck downward a lot." -P13*

Also, for 3 of the participants, the arm menu felt shorter in a virtual environment: *"It felt that the arm was shorter in VR causing the arm menu to feel like it was at my shoulder." -P18*

In the case of the ray-casting selection technique with the arm placement menu, the participant's primary hand had to be stretched every time they needed to select the menu item that resulted in a bad experience: *"The worst experience is with arm and controller because painting a laser on to arm is difficult." -P14*

For the hand graphical menu, the participants preferred this menu type for games in virtual environments: *"I think that for VR games the hand placement would be the best since it's relatively small and easy to use." -P10*

Furthermore, the users put the waist graphical menu into the same category with the arm menu in terms of its difficulty: *"When using the menus placed on the waist, the user has to look down which may cause stain on neck after prolonged use." -P15*

Overall, the participants found the spatial menus easy to use, significantly less physically and mentally demanding in comparison with other graphical menus: *"I preferred the spatial menus as they were the easiest to select using all three selection methods. It required the least amount of movement which I liked." -P3*

Notwithstanding, 14 participants also reported the eye gaze as their favorite selection technique: *"The easiest and fastest method of selecting a menu was the eye gaze. I did not need to spend any time aiming my head or the controller to the menu item I wanted to select." -P15*

However, for 5 of them, the eye gaze had issues with accuracy: *"Eye gaze sometimes was not accurate, I had to look at the above menu item to get the one below. It only applies to middle menu options." -P4*

Below, we discuss the implications of our findings and provide design recommendations for implementing body-referenced graphical menus in virtual environments.

Table 3: Results on Friedman's test and post-hoc analysis for Likert scale data.

| # | Friedman's Test | Spatial vs Arm | Spatial vs Hand | Spatial vs Waist | Arm vs Hand | Arm vs Waist | Hand vs Waist |
|---|---|---|---|---|---|---|---|
| Q1 | $\chi^2(3) = 39.701, p < .0005$ | $Z = -4.215, p < .001$ | $Z = -3.58, p < .001$ | $Z = -4.247, p < .001$ | $Z = -2.728, p < .01$ | $Z = -0.23, p = .818$ | $Z = -2.183, p < .05$ |
| Q2 | $\chi^2(3) = 24.019, p < .0005$ | $Z = -3.53, p < .001$ | $Z = -2.922, p < .01$ | $Z = -3.367, p < .01$ | $Z = -2.402, p < .05$ | $Z = -1.93, p = .054$ | $Z = -0.956, p = .339$ |
| Q3 | $\chi^2(3) = 44.005, p < .0005$ | $Z = -4.215, p < .001$ | $Z = -3.95, p < .001$ | $Z = -3.841, p < .001$ | $Z = -3.292, p < .01$ | $Z = -2.372, p < .05$ | $Z = -0.951, p = .341$ |
| Q4 | $\chi^2(3) = 28.253, p < .0005$ | $Z = -3.863, p < .001$ | $Z = -3.093, p < .01$ | $Z = -3.211, p < .01$ | $Z = -2.043, p < .05$ | $Z = -1.919, p = .055$ | $Z = -0.106, p = .915$ |
| Q5 | $\chi^2(3) = 3.48, p = .323$ | $Z = -1.663, p = .102$ | $Z = -1.511, p = .131$ | $Z = -1.414, p = .157$ | $Z = -0.368, p = .713$ | $Z = -0.552, p = .581$ | $Z = -0.184, p = .854$ |
| Q6 | $\chi^2(3) = 35.645, p < .0005$ | $Z = -4.035, p < .001$ | $Z = -3.46, p < .01$ | $Z = -3.552, p < .001$ | $Z = -3.031, p < .01$ | $Z = -3.142, p < .01$ | $Z = -0.15, p = .881$ |
| Q7 | $\chi^2(3) = 43.882, p < .0005$ | $Z = -4.121, p < .001$ | $Z = -3.562, p < .001$ | $Z = -3.777, p < .001$ | $Z = -3.198, p < .01$ | $Z = -3.283, p < .01$ | $Z = -0.655, p = .513$ |

## 5 DISCUSSION

### 5.1 Menu Placements

*Placing a Graphical Menu on an Arm.* Our experiment indicates that the arm menu placement took a significantly longer time to complete the menu task in comparison with the spatial, hand, and waist placement menus. This is primarily because the user needed to adjust their arm position in order to select the corresponding menu items, and the system message was in a fixed position but located further compared, for example, to the spatial menu. Additionally, the user needed to turn their head up and down to look at the system message that showed them what book to choose from the menu. Overall, the arm graphical menu was the least favored among participants. The participants found this type of menu hard to use (physically and mentally), were highly frustrated while completing the menu tasks, and felt that the menu was in an uncomfortable position. This is primarily because we used an off-the-shelf HTC Vive implementation and tracked the body using two controllers and one additional tracker attached to a participant's waist without any additional custom hardware. On one hand, this approach reflects real-world usage but, on the other hand, does not give that high accuracy. However, we foresee better full-body tracking accuracy to make the body placements feel more intuitive and natural. For example, Caserman et al. [8] presented an accurate, low-latency body tracking approach using a Vive headset and trackers that can be incorporated by VR developers "to create an immersive VR experience, by animating the motions of the avatar as smoothly, rapidly and as accurately as possible."

For the arm menu with a linear shape, the participants reported that it was hard to reach the menu items that were placed closer to the elbow. Our quantitative results also indicate that linear and radial shapes of the arm placement menus are significantly slower than the linear or radial spatial graphical menus. The majority of the participants noted that the arm menu in conjunction with the head or ray-casting felt awkward and difficult. In the case of the head input, the participants had to turn their heads up and down a lot of times and keep adjusting their arm. Overall, the participants concluded that the arm placement technique felt more intuitive when it was combined with a radial shape or linear shape and eye gaze selection technique. Interestingly, even though the participants preferred eye gaze for the arm placement, this selection technique took a higher number of target re-entries for completing menu tasks than the arm head selection technique. Thus, we suggest implementing an arm placement menu with any menu shape and eye gaze selection technique as it provides a more natural and physically easier interaction. Additionally, we found that for the arm graphical menu, it is strongly recommended to shorten the amount of interaction, because a prolonged duration causes arm strain, especially, when the arm has to be held up.

*Attaching a Graphical Menu to a Waist.* We found that the waist menu was significantly faster than the arm graphical menu. The participants found this menu physically hard and difficult to use. Also, the majority of the participants reported that the waist menu in conjunction with the head input would cause neck strain. Therefore, we highly recommend placing the system messages near the waist graphical menu and to not use it for a prolonged duration. For the waist menus, the eye gaze selection technique usually had a higher number of target re-entries than the head or ray-casting, and the head selection technique had a higher average completion time than ray-casting. We find that the best interaction technique for the waist menus is the linear shape (based on the participants' preference) with the ray-casting selection technique.

*Placing a Graphical Menu on a Hand.* The hand menu was the participant's second favorite graphical menu. We found that for the hand menu, task completion time is significantly faster than for the arm menu, but it is also more prone to errors than the arm placement menu. Additionally, we found that the head selection takes more time for the hand placement menu than for the ray-casting technique. Further, eye gaze has a significant number of target re-entries than ray-casting. Even though 4 of the participants reported the hand graphical menu as their favorite, others found it is somewhat hard and tricky to use. We believe this is primarily because the participants had to hold up their hands and adjust its position in order to choose the menu items. Overall, based on the feedback and results from the participants, we suggest using the hand menu in conjunction with a radial shape (as it was mostly favored by the participants but did not have any issues with other quantitative metrics) and the ray-casting selection technique.

*Placing a Graphical Menu in the Virtual World.* Overall, the spatial graphical menu was the most favored among participants. The spatial menu was only significantly faster to use in comparison with the arm menu. The users noted that the spatial menu would be better to use with a radial shape and ray-casting or eye gaze selection techniques. Even though this placement menu was participants' overall best graphical menu to use, for the spatial menus, we found that the head selection took more time to complete a menu task and the eye gaze menu had more target re-entries than ray-casting. Therefore, we suggest using the spatial graphical menu with ray-casting and any menu shapes.

### 5.2 Menu Selection Techniques and Shapes

For the selection techniques, we found the participants made significantly more errors when selecting the menu items with eye gaze. Likewise, the number of target re-entries was significantly higher in the case of eye gaze than with the ray-casting or head selection techniques. This is primarily because eye-tracking technology is highly sensitive to eye movement making the user accidentally select the wrong menu items and leave the target menu item and then go again inside the target significantly more often. In the future, we foresee better eye-tracking accuracy that will make the eye gaze selection technique more intuitive and accurate to use [23].

The ray-casting selection technique was faster than the head selection technique. However, we found that the participants made more errors when selecting the menu items with ray-casting than with the head selection. Likewise, the number of target re-entries was higher in the case of ray-casting than with the head selection technique. This is primarily because the ray-casting input does not require the user to precisely select the menu items and adjust the head position (which makes the head selection more time-consuming). Overall, ray-casting felt intuitive and easy to use for the participants. Moreover, the laser pointer of ray-casting gave the user additional control over the graphical menus.

The head selection technique was the participant's least favorite. We found that the head input took more time in comparison with the ray-casting selection technique because each participant had to adjust the head movement in order to select the menu item and be more accurate and precise. This is also the reason why the head input is less prone to errors and number of target re-entries. The participants reported that the head controls did not have a very good place on any graphical menu.

Overall, shapes did not matter significantly for the participants. Also, we did not find any significant difference in task completion time, error rates, or number of target re-entries which resonates with a finding of Santos at al. [22] that suggests that the user experience is not greatly affected by linear or radial menu shapes.

Based on the results of our experiment, we were unable to accept the **H1** and **H2** hypotheses. We did not find any significant difference in task completion time, error rates, and number of target

re-entries for the spatial graphical menu with the radial shape and the ray-casting selection technique in comparison with the other graphical menus. Moreover, even though the participants preferred a more conventional spatial menu, shapes and selection did not matter for them, as they selected all two menu shapes (linear and radial) menu shapes and all three selection techniques (ray-casting, head, and eye gaze) as their preferred for the spatial menu.

## 6 LIMITATIONS AND FUTURE WORK

There are a few factors that could have affected our results. In particular, 2 participants had difficulty with eye-tracking technology. We noticed that those participants wore thick lenses that made eye-tracking less accurate at recognizing eye movement. Additionally, for the arm placement menu, the users had to rotate the arm accordingly before being able to even see or select items from the menu which made the arm graphical menu less comfortable. Ideally, the arm menu should be more flexible and intuitive to use with a better design approach. Also, the participants reported that they wanted their virtual arm to be longer. We believe that with better full-body tracking technology (e.g., by using additional custom hardware), we can solve this issue and match the real user's arm with a virtual arm. Further, the system message was placed in the center near the virtual object in the VR environment and was not attached to a placement menu (e.g., near the arm placement menu). Ideally, the system message should have the instruction panel close to each type of menu to homogenize the distance from the instructions to the target. We also did not implement additional pointer smoothing or padding between control elements that could help participants make less errors. Moreover, the text was sometimes unclear to read when appeared on the participant's hand or arm. We think this is primarily because of the headset resolution that can be solved with a high-resolution VR headset.

Given a variety of criteria that need to be considered when implementing graphical menus in VEs, in the future, it would be interesting to investigate how various menu hierarchical depths or sizes and other types of graphical menus (e.g., object-referenced or device-referenced) affect the task completion time, error rates, number of target re-entries, and other quantitative and qualitative metrics. Additionally, even though Bowman et al. [5] specified that "body-centered menus also do not inherently support a hierarchy of menu items", it would be interesting to see how depths/number of menu elements indeed affect such menus. In general, our study was developed to account for performing menu tasks in a static standing context, however, in the future, we would like to investigate how body-referenced menus can be applied for more dynamic virtual environments.

## 7 CONCLUSION

We presented an in-depth systematic study on evaluation of body-referenced graphical menus in virtual environments in terms of different placements (spatial, arm, hand, and waist), menu shapes (linear and radial), and selection techniques (ray-casting with a controller device, head, and eye gaze). Our results show that the spatial, hand, and waist menus are significantly faster than the arm menus. Moreover, we found that the eye gaze selection technique is more prone to errors and has a significantly higher number of target re-entries than the other selection techniques, however, we did not find any significant difference in task completion time, error rates, and number of target re-entries for the menu shapes. We found that a significantly higher number of participants ranked the spatial graphical menus as their favorite menu placement and the arm menu as their least favorite one. We also provided design

guidelines and recommendations for body-referenced graphical menus including its preferred shapes and selection techniques.

## ACKNOWLEDGMENTS

This work is supported in part by NSF Award IIS-1638060 and Army RDECOM Award W911QX13C0052. We also thank the anonymous reviewers for their insightful feedback. We are further grateful to the Interactive Systems and User Experience Lab at UCF for their support.

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
