# OpenReview forum: "Evaluation of Body-Referenced Graphical Menus in Virtual Environments "
_graphicsinterface.org/Graphics_Interface/2020/Conference — GI 2020_

### Official Review · AnonReviewer2 · 2020-04-21
**Good study paper, with in-depth analysis**

**Rating:** 6
**Confidence:** 3

**Review:**

Good study paper, with in-depth analysis

This paper presented a comprehensive experiment in VR, focusing on various types of menu placements, shapes, and selection techniques. Both quantitative results (incl. task completion time, error rates, # of re-entries) and post-study comments were reported.

Pros:

I enjoyed reading this paper. The topic domain was nicely chosen -- I agree that this work made a contribution by analyzing the experimental results and proposing categorized recommendations for developers in this field. The discussions of each independent variables and their interactions were inspiring to read.

Cons:

As for the clarity of presentation, I have a few concerns:
	• In Section 3.4, it was mentioned that the participants got instructions "in a system message". But where is this system message displayed? In the 4 different menu placement conditions (arm/hand/waist/spatial), obviously the target UI element was displayed at different locations. Then it would be important to know where the participant got the instruction, because this would have an effect on the task completion time.
	• Since Menu Placement is a variable with 4 levels (within-subject), and there were 24 participants in total, the degrees of freedom for F values before sphericity correction shall be (3, 69) instead of (3, 23), isn't it? Please also check the degrees of freedom reported in other parts of this paper.
	• For Fig. 3: The legends and x-axis groups could be re-designed to better convey the message illustrated in Section 4.2. Now the figure focused more on the comparison of menu placements for each type of selection technique, rather than comparing the selection techniques.

In summary, I would like to accept this paper, and I am looking forward to seeing your answers to these questions.

---

### Official Review · AnonReviewer3 · 2020-04-21
**I think the paper should be accepted with some revisions. Paper is well motivated, researched, and key contributions are clearly laid out and explained. There are a few elements of concern, including the body-placement implementations lacking in robustness. As well as the results section being overly dense, since the charts are used nearly exclusively for preference ratings but only tables for calculations.  I’d consider adding a chart or two to compare timing, errors, or some other value.**

**Rating:** 6
**Confidence:** 5

**Review:**

Motivation:
1. Solidly written motivation and well-cited literature review.
2. The authors seem to provide a strong justification and explanation for exploring optimal interaction experiences for placement of graphical menus on body parts in VR body.
3. While there has been extensive studies on menu selection in VR, I am not familiar with one that is specific to menus located on virtual body parts. The proposed work seems to have at least reasonable novelty in this regard.
4. The proposed work of study on VR body-referenced menus seems to be an appropriate fit to the conference's focus on graphics and on interactions.
5. Hypotheses are solid and provide a good frame for the rest of the study
6. One of  the concerns with the motivation is the fact that the authors found body-referenced graphical menus to be insufficiently explored. However, I found the implementation of body-referenced options in the paper’s methodology to be lacking in a few key ways (see Approach section).
7. It’s possible that the lack of body-referenced graphical menu studies is due to the technological limits. This factor does not seem to be explored in the evaluation sections beyond “VR technology will get better in the future”.
8. The proposed work's contributions listed at the end of page 1 seem valid, but could be better condensed. That is, the two bulleted items are most redundant since most of the text is repeated except for the first word. Perhaps revise the text so that the two bulleted items are instead more succinct, such as merging them into a single merged sentence.
9. The authors were very thorough in citing and discussing references that strongly covered domains that both directly and peripherally related to the proposed work. It is clear that the authors put a lot of effort and planning to provide a detailed literature review and discuss how they differ or support the authors' proposed work.
10. Although the authors were quite thorough with their 25 references, one concern was that only one reference was from the past year and only three references were from the past two years. That is, most of the references were seminal but not state-of-the-art. If space permits, I recommend that the authors explore and include more recent related references of importance, and discuss why those recent works still do not address the challenges that remain open to the authors.
11. Suggested References:
Monteiro et al. Comparison of Radial and Panel Menus in Virtual Reality. IEEE Access. 2019.
Park et al. HandPoseMenu: Hand Posture-Based Virtual Menus for Changing Interaction Mode in 3D Space. ACM ISS. 2019

Approach:
1. Study design is well-written, well-motivated, and the questionnaire adequately captures the user experience
2. The three metrics of task completion times, error rates, and number of target re-entries seem appropriate and informed given the length of the work, and were also appropriately discussed.
3. I particularly appreciate the attention put into whether participant’s short-answer feedback matches with that of the usage metrics. For example, I like that the paper points out differences between what users think “feels” better for them despite lower accuracy and/or higher time taken.
4. Several references to “radial menus” were made but the one figure in the paper seems to depict linear menu placements. Radial menus should be shown in a second figure, especially since the paper is primarily UI-focused
5. Paper only explores a completely flat menu hierarchy. While exploring multiple hierarchies and different forms of input (e.g. drop-down menus) could be understandably out of scope, the authors should still mention justification on using the most basic of menu designs. The chief concern here is that most menus are not simple “pick from list” single-tier affairs, and that could make the paper’s findings not readily applicable to VR UI development.
6. My main concerns on the approach is the robustness of the system. Despite the fact that the paper’s main motivation is in body-referenced menus, the implementation does not integrate any kind of body-tracking.
-The effects of this are seen primarily in the arm-menu. Many participants reported the “VR arm” did not match their own; they felt it was too short, long, or was not in the location they anticipated relative to their head.
-The hardware used was an HTC-vive, which only directly tracks hand movement through the location and orientation of the controllers. This study then extrapolates an arm, but it appears this implementation was insufficient for several participants.
-Perhaps the system did not account for differences in height, which directly correlates to forearm length
If this problem can be solved via software, it did not appear that the research study took this issue into account or attempted to solve it. If the problem can only be solved via hardware, the study did not appear to integrate body-tracking hardware to bridge this accuracy gap.
-This does not invalidate the research study. In fact, it’s easy to argue that using an off-the-shelf HTC Vive implementation without custom hardware reflects real-world usage better. However, I would like to see that explicitly laid out in the Discussion section. I don’t think deferring to “better full-body tracking technology in the future” quite captures the level of discussion needed for this particular point.
7. Input errors could potentially be due to software implementation. It’s unclear whether the authors explored pointer smoothing (for ray-casting and eye-tracking), padding between control elements, a buffer for pointer leniency (so that the pointer does not have to be exactly inside the button).
-If these were not implemented, could this be a confounder? This should be discussed in the Discussion or Limitations section
8. One mystery about the paper was in regards to the orientation and direction of the body parts. That is, does the study take into account different orientations of the limbs such as the rotation of the parts, the facing direction of the hands, and the angle of the arms?
9. What was the starting position of the limbs for each stage? Were the users' limbs at rest on their side and were instructed to position them to reveal the menus, or were the users' limbs always in the necessary starting position to trigger the menu display? This information seemed to have been skipped over.
10. How were the menus triggered? Were the limbs recognized and the menus automatically placed on them? This information may be useful to readers who may not be familiar with how menus work in VR given the context of body parts. The only discussion was brief mentions of full-body tracking in the latter discussion sections, when the paper would benefit from earlier mention in the study approach's sections.
11. How was the amount of menu content for placement on the limbs determined? The approach described states a total of six, but how was that number decided? I believe that content amount in the menus may cause an effect, but it seemed like the menu content was arbitrarily selected or lacked justification?
12. One of the major limitations of the approach is the singular task of menu interactions in terms of selecting books. This is not a detriment to the study since this task does appear to cover quite a number of similar VR interactions. However, I was curious to know how generalizable this type of study is for VR menu selection in general. I believe that the study is appropriate for static standing context to perform precise selection of options, but I am unsure if this type of interaction is appropriate for more dynamic VR interactions. That is, maybe more dynamic VR interactions may prefer more immersive experiences where the user is okay with performing more physical actions to view VR menus on their body parts for more "cinematic" experiences. Perhaps optimal menu selections may be different when the user is more actively moving. It is hard to tell on this type of generalizability with the one task that was conducted. The authors may want to consider revising their approach to reflect this more constrained interaction scenario---which is still important---so that readers are not misled by the paper's actual evaluation. Alternatively, the authors should defend why their interaction scenario is actually generalizable. With the paper's current state, this question remains unaddressed.

Evaluation:
1.This is probably personal preference, but I felt most of the text reporting completion times and error rates could be much more succinctly summarized in a table or integrated into Figs 2, 3, and 4. Doing this would likely free up space for additional UI figures that are much needed.
2.The evaluation provided detailed quantitative and qualitative outcomes that were appropriate for the study that they conducted.
3. It is general practice to remove the leading zero before the decimal point when listing p-values, since these leading zeroes are redundant. Please remove leading zeros in your listed p-values in the paper.
4. Table 2 presents the data with detailed results, however the table header values (i.e., TCT, ERR, TRE) are not too intuitive. I suggest replacing these initials with more intuitive names, such as Completion, Error, and Re-Entries.
5. One discussion area that I was interested in knowing about but did not see was the authors' observations and the participants' remarks regarding how they posed their body parts to prompt and view the menus. Specifically, I was curious if participants were consistent in how they posed their body parts or if people biased their body parts uniquely from other participants. The only remarks that I saw was on isolated observations that were consequences of participants failing to trigger them in the latter discussion sections.
6. The results that were provided show valid and relevant statistical significance that does highlight which menu interactions were better than others given the context of the study.

---

### Official Review · AnonReviewer1 · 2020-04-21
**Flaws in study design need to be addressed**

**Rating:** 5
**Confidence:** 5

**Review:**

Evaluation of Body-Referenced Graphical Menus in Virtual Environments

In this paper, the authors present a study to evaluate different aspects of menu selection task in Virtual Reality (VR). Four different types of menu placement locations, two menu shapes, and three menu selection techniques were evaluated. Overall, the paper is well motivated and the study design attempts to answer key questions about menu design in VR. The study and the associated discussion of the findings are the key contributions of this paper. The study design, with 24 conditions, is quite extensive and the stats are appropriate.

I have some concerns about the study design and the thoroughness and clarity of describing the study setup. I think these concerns need to be address to help contextualize the results better and improve the readability and replicability of the study.

Primary concern:
- The primary issue is the design of the task itself. Each trial goes something like this: the rectangular panel in the centre of the viewer's field of view displays the next target (instruction panel). The participant then either clicks the 'start' button or goes ahead and selects the indicated menu item. ( It is unclear whether the participant has to press start and then press the requested menu item.) Nonetheless, we can assume that for atleast eye tracking and head movement conditions the pointer travel distance from the instruction panel to the menu varies for different menu placement locations. For example, the menu placed on the arm is quite far away compared to the spatial menu. Based on Fitts' law, it is not surprising that the arm placement fared the worst in task completion times. Since the dependent variable is task completion time, the distance travelled from the pointer resting position to the target menu position can affect the study significantly. This implies that some of the conclusions drawn at the end of the paper would need to be updated.  Ideally, the study design should have attached the instruction panel close to each type of menu to homogenize the distance from the instructions to the target. Alternatively, the paper has to acknowledge this confounding error and revise the findings.

- Following up from the above point, I want to note that the description of different conditions could be improved through detailed explanation and visuals. For example, was the laser selection pointer and the menu selection visually displayed for all conditions? if so, how? The inserted image only shows visuals for the ray-tracing condition. Images are needed for all the task conditions in the study. Without this, there are several lingering assumptions on how people may have actually used the different menu selection techniques.

Other minor concerns:
- Actual task for the study is unclear from the study design and procedure. For example, do the participants click on the 'start' before making the next menu selection? It appears so from the images. But, this has to be explicitly mentioned in the procedure. Also, were the participants seated or could they be standing and/or walking around?

- How were the hypotheses reached? Although some of this is implied from the related works section, it is better to repeat the information in the study hypothesis.

Overall, I think the study findings could be improved by fixing the issues pointed out.

---

### Meta-Review · Area_Chair1 · 2020-04-24

**Recommendation:** Accept
**Confidence:** 4

**Metareview:**

Overall, all the reviewers acknowledge that the paper is well motivated (R1,R3) with strong justification (R3), and a comprehensive experiment (R1,R2). R3 provides a thorough overview of the key strengths and issues with the paper. R1 and R2 raise specific concerns about aspects of the study design.

I'm highlighting some of the key concerns here. Firstly there are questions about the placement of the instruction/system message (R1,R2). R1 gives a detailed explanation of why the design of the system message and its placement needs to be clarified and the discussion section updated. R3 asks for addresses the issues related to the robustness of body tracking. R1 and R3 also request several key pieces of information related to the study task. R3 suggests commenting on the generalizability of the study tasks. R2 suggests checking the degrees of freedom for some of the reported F values.

R2 & R3 suggest edits for the figures to improve clarity. R1 & R3 request including images for different task conditions for clarity.

Overall, I think the paper can be accepted with minor revisions addressing the issues raised by the three reviewers.

---

### Decision · Program_Chairs · 2020-04-25

Accept